# Heat Shock Proteins (Hsps) in Cellular Homeostasis: A Promising Tool for Health Management in Crustacean Aquaculture

**DOI:** 10.3390/life12111777

**Published:** 2022-11-03

**Authors:** Vikash Kumar, Suvra Roy, Bijay Kumar Behera, Basanta Kumar Das

**Affiliations:** Aquatic Environmental Biotechnology and Nanotechnology (AEBN) Division, ICAR-Central Inland Fisheries Research Institute (CIFRI), Barrackpore 700120, India

**Keywords:** heat shock proteins, crustaceans, protein homeostasis, protective immunity, both abiotic, biotic stresses

## Abstract

Heat shock proteins (Hsps) are a family of ubiquitously expressed stress proteins and extrinsic chaperones that are required for viability and cell growth in all living organisms. These proteins are highly conserved and produced in all cellular organisms when exposed to stress. Hsps play a significant role in protein synthesis and homeostasis, as well as in the maintenance of overall health in crustaceans against various internal and external environmental stresses. Recent reports have suggested that enhancing in vivo Hsp levels via non-lethal heat shock, exogenous Hsps, or plant-based compounds, could be a promising strategy used to develop protective immunity in crustaceans against both abiotic and biotic stresses. Hence, Hsps as the agent of being an immune booster and increasing disease resistance will present a significant advancement in reducing stressful conditions in the aquaculture system.

## 1. Introduction

From the past to the present, crustacean aquaculture has been increasing considerably in order to reach the higher food demands of the world. The total crustacean aquaculture production, from over 30 different species, was 8.4 MT valued at USD 61.06 billion, with an average annual growth rate of 9.92% per year since 2000 [1,2]. The marine shrimp currently dominate crustacean aquaculture at 5.51 MT or 65.3% of total crustaceans (valued at USD 34.2 billion), followed by freshwater crustaceans (2.53 MT or 29.9% of total crustaceans and are valued at USD 24.3 billion). Despite tremendous development, the aquaculture sector is confronted with various problems, some of which have a detrimental impact on its ability to achieve sustainable outcomes [3,4]. Disease outbreaks in aquaculture have been one of the most serious issues, causing significant output losses, sometimes up to 100%. The adverse environmental conditions in the culture systems caused by, for instance, extreme weather events and crowding stress are some of the major drivers of disease emergence. Pathogenic and non-pathogenic microbial populations coexist with farmed animals in an aquaculture production system, however, when they are stressed, acute disease and infections may develop [5,6]. It is noteworthy that not all pathogens cause significant mortality; some contribute to heavy losses through their negative impacts on growth, appetite, susceptibility to adverse environmental conditions, and many others. For instance, acute hepatopancreatic necrosis disease (AHPND) caused by *Vibrio parahaemolyticus* and Microsporidian *Enterocytozoon hepatopenaei*, a shrimp pathogen that was discovered in Thailand over ten years back, has now become a widespread and highly impacting pathogen responsible for growth retardation, yet without causing a significant effect on shrimp survival [7,8,9]. Based on the discussion made above and the evidence from earlier studies, it can be suggested that biological agents/pathogens and environmental factors interrelate in various complex ways with the host in order to cause diseases.

The culture of crustacean animals in the most optimal conditions is not often economically feasible; as such, there will always be a risk of infection and a need for effective disease control strategies in order to make the aquaculture industry more sustainable. Among the various drugs, antibiotics are the most popular and widely used chemotherapeutic agent used to control bacterial disease in crustacean aquaculture [10,11]. However, the wide use of antibiotics, and especially their application at subtherapeutic doses, inevitably results in the development of antibiotic-resistant bacteria. In fact, when antibiotics are used, the resistant bacterial strains can multiply rapidly due to the fact that their competitors (sensitive bacterial strains) are suppressed. Moreover, it is generally accepted that the acquisition of resistance is more rapid against bactericidal antibiotics. As antibiotic-resistant genes are often located on mobile genetic elements, such as in transferable plasmids and integrons, the antibiotic-resistant mobile genetic elements can be transmitted via horizontal gene transfer, not only to other bacteria but also to terrestrial bacteria, including humans and animals’ pathogens [12,13,14]. This resistance has already been reported in some human pathogenic bacteria, including *Salmonella enterica* serotype *Typhimurium* and *Vibrio cholera*. Given the worldwide trade in aquaculture products, health problems associated with antibiotic use are not limited to aquaculture food-producing countries but are in the countries importing antibiotic-treated food products [15,16]. Due to the magnitude of the challenges that the industry is faced with as it grows, new alternatives to the use of antimicrobials are urgently needed, since no anti-infective treatment appears to be capable of solving every problem. 

Crustaceans, similar to other invertebrates, lack an adaptive immune system. Instead, they rely on their effective cellular and humoral innate immune system in order to defend against hostile microorganisms. In crustaceans, including shrimps, the major immune reactions occur in the hemolymph, which consists of three different principal types of hemocytes, i.e., hyaline, granular, and semi-granular hemocytes [17,18]. Over the past few years, several strategies/preventive measures to enhance non-specific immune defense systems and protect crustacean animals from pathogenic microbes, without using antibiotics, have been developed; further, some of them have even been applied successfully. For example, in regard to the monitoring and managing of microbes in the culture system, some of the promising alternatives are microbe-associated feeding strategies, (i.e., natural or plant-based compounds), prebiotics, probiotics, and synbiotics, improving welfare and stress prevention, and the use of natural immunostimulants. Interestingly, the heat shock proteins (Hsps) that have been reported to potentiate the generation of pro-oxidant activity are responsible for the induction of protective immune responses in crustacean species [19,20]. Hsps, also known as stress proteins and extrinsic chaperones, are a group of evolutionarily conserved proteins. Hsps are well known as molecular chaperones, and they are known for aiding nascent polypeptide folding and oligomerization; protecting proteins from irreversible denaturation; re-folding or degrading damaged proteins; translocating proteins into membrane-bound cell compartments; and contributing to disease resistance [19,20,21,22,23,24,25]. Hence, a holistic approach—which includes the environment, host, and pathogen—and develops pharmacological agents capable of inducing Hsp responses, could protect the animals at different stages of life and against a broad spectrum of infectious agents; further, this type of approach will also most likely be the most sustainable [26,27,28,29,30]. In this new concept of disease control strategy, measures that prevent disease are the most important health management options. The focus of the present review is to provide a details overview of Hsps and their potential role in host immune response, protein homeostasis, and cross-protection against abiotic and biotic stress through different regulatory pathways. In addition, we have also tried to explore the in vivo working mechanism of Hsps by using a model organism, *A. franciscana* (brine shrimp). This model is utilized through different regulatory pathways and possibilities for the purposes of developing compounds/molecules (e.g., plant-based compounds) that can induce the production of Hsps within the host, as well as confer protection against various abiotic and biotic stresses. 

## 2. Regulation of the Heat Shock Protein Response 

Heat shock proteins (Hsps) are evolutionarily ancient and extremely conserved proteins found in almost all living organisms, ranging from archaebacteria, prokaryotes, and eukaryotes. Hsps play an essential role in regulating cellular metabolism in stressful conditions. These proteins are present in several intracellular locations, e.g., in the nucleus, mitochondria, endoplasmic reticulum (ER), and chloroplast and cytosol of eukaryotes [31,32,33]. In general, Hsps are classified into several families based on their function, molecular mass, and sequence homology (i.e., Hsp60, Hsp70, Hsp90, and sHsps) (Figure 1) and also, they can be grouped according to their nearest size family (e.g., Hsp84, Hsp85, and Hsp86 in the Hsp90 kDa family) [34]. 

Hsps plays a key role in cellular processes that occur during and after exposure to oxidative stress that is caused by hazardous environmental and/or microbial agents. As a result, the normal intracellular reducing environment is compromised, which leads to oxidation and aggregation of key proteins and DNA, ultimately resulting in cellular dysfunction. Due to their versatile functions, Hsps can intervene following oxidative stress at several levels [18,19,35]. Firstly, some Hsps, mainly the Hsp70 family members, play a crucial role in protein sorting and quality control via selecting and directing abnormal proteins to the proteasome or lysosomes for degradation; thus, Hsps aid the clearance of damaged proteins [36]. In some cases, where misfolded proteins need to be rescued, the same machinery facilitates the correct folding of damaged proteins. Moreover, the Hsp families, such as Hsp27, Hsp70, and Hsp90, can negatively regulate apoptosis via the binding and inhibiting of members of the apoptotic cascade. Some Hsps have immuno-enhancing actions (for details, see the section on Hsps and immunity). Among the different Hsp families, Hsp70 is the largest and most highly conserved of the stress protein families. At least 121 proteins have been characterized within this family, and cross-hybridization occurs across various species, such as in mammals, fish, and mollusks [37,38,39]. 

After the discovery of heat shock proteins, it was clear that the heat shock protein response requires a specific transcription factor [40,41,42,43,44]. In crustaceans, a few reports have suggested that the heat shock transcription factor plays a significant role in determining the heat shock response and also in tolerance against stressful conditions. Tan and Macrae [45] reported that the heat shock factor 1 (Hsf1) transcription factor induces stress tolerance (drying) in diapausing brine shrimp cysts and is responsible for the improved growth and survival of *A. franciscana* [45]. In another study, Sornchuer et al. [46] demonstrated that Hsf1 has an important role in the thermal stress response and regulates the transcription of heat shock proteins and immune-related genes in *P. monodon* [46]. However, we still need to carry out detailed investigations and obtain more information in order to characterize the mechanism of heat shock response regulation in crustaceans. Moreover, in eukaryotes, the heat shock response is well documented, and this study shows that the induction of Hsp transcription is mediated by a pre-existing transcription factor, i.e., the heat shock factor or the heat shock transcription factor (HSF) [47]. The HSF upon activation binds to the heat shock element (HSE) at the promoter region (5’ upstream end) of the Hsp gene and induces the transcription of Hsp [48]. The binding motif of the HSE is composed of nGAAn blocks (5 bp) in alternate orientation, and for the stable binding of the HSF and the HSE, at least three units are required [49]. In prokaryotes (e.g., *E. coli*), the σ32 regulatory protein is responsible for Hsp expression [50]. Additionally, as an alternate subunit of the bacterial RNA polymerase, σ32 replaces the σ70 normal regulatory proteins during heat stress [51]. HSF1 and σ32 share basic mechanistic properties, however their structure or sequence are not related; further, it has been found that protein homeostasis disturbance results in the activation of hsf1 and σ32 [52]. 

HSF genes have been reported from yeast (*Saccharomyces cerevisiae*), plants (*Arabidopsis* sp.), fruit flies (*Drosophila* sp.), chicken (*Gallus* sp.), *A. franciscana*, and *P. monodon* [53,54,55,56,57]. Sequence comparison of the HSF gene from different species showed that DNA binding and the oligomerization domain is strongly conserved. The HSF contains two highly conserved regions, i.e., a ~100 amino acids NH2-terminal DNA binding domain and an adjacent trimerization domain having 3 hydrophobic heptad repeats, leu zippers [33]. The activation of the HSF involves cellular factors as the intermediary sensors that regulate the activity of the HSF during non-stressful conditions [58]. In animals, the HSF is maintained as a monomeric form through transient interaction with Hsps. However, during stressful conditions, the HSF was released and formed trimers with the HSE, which resulted in the induction of Hsps. Subsequently, Hsps bind with denatured/misfolded protein aggregates and maintain the homeostatic condition in the cells [59,60,61,62,63]. 

Based on previous studies, a hypothetical illustration of the possible methods involved in the activation of the heat shock protein response, and its potential role in maintaining homeostasis and host health, is depicted in Figure 2. Briefly, the stressful stimuli, heat-shock-protein-inducing (Hspi) conditions (e.g., environmental, pathological, and physiological) [64,65], or compounds (e.g., plant-based, natural polymers, etc.) [66,67] develop oxidative status inside the host. Subsequently, it facilitates the phosphorylation and nuclear translocation of the HSF (in the native form present in the non-phosphorylated form attached with heat shock proteins such as Hsp90) [68] where (inside the hemocyte nucleus) the HSF binds with the HSE at the promoter region (5’ upstream end) of the Hsp gene and induces the transcription of the Hsp gene and production of Hsps [69,70]. The Hsp (e.g., Hsp70) functions as a non-covalently molecular chaperone bind with a hydrophobic exposed segment of unfolded proteins [71,72]. Further, this prevents the aggregation of inappropriate or unfolded proteins; inhibits the misfolding of the polypeptides; transports immature polypeptides to target organelle for the purposes of final packaging and repair, as well as denaturation or degradation of misfolded proteins through proteasomes or lysosomes (proteolysis), which cannot be repaired [73]; and, also, maintains protein homeostasis. The extracellular Hsp also functions as a chaperokine and binds with Toll-like receptors (e.g., TLR-4), which are expressed on hemocytes that lead to the maturation and activation of hemocytes [74,75,76]. The activated hemocytes induce the production of astakines that binds with the binding site of a pathogen in the host cell and decrease the chance of the pathogen attaching to its host cells [77,78,79]. The hemocytes (activated by Hsps) also induce the expression of transglutaminase (Tgase) [80], which form a blood clot in the presence of Ca^2+^ and prevents the spread of pathogens and increases the expression of antimicrobial peptides (e.g., crustin, lysozyme, etc.) that has bactericidal activity against Gram-positive and Gram-negative bacteria [81,82,83,84]. The p38 (mitogen-activated protein kinase, MAPK) activated by hemocytes, increases the expression of antimicrobial peptides (e.g., crustin, lysozyme, etc.) and has a critical role in defense against bacterial and viral infection [85,86,87,88,89]. Hemocytes activate the ProPO cascade in the presence of Ca^2+^, leading to melanization and further killing of the pathogens [90,91,92,93].

## 3. Factors Modulate Heat Shock Protein Response

The induction of Hsps in response to cellular stressors was initially considered a short-term functional response, with a range of essential housekeeping and cytoprotective functions. These stressors, in general, induce protein damage and increase the susceptibility of host animals further to subsequent stressful conditions. However, accumulating pieces of evidence, over the past few decades, have suggested that Hsps play a significant role in the regulation of the immune response in invertebrates (Figure 3 and Table 1). The heat shock protein expressions influenced by either abiotic or biotic stresses are summarized in the section below.

### 3.1. Environmental or Abiotic Stresses

The physiological status of crustaceans is greatly influenced by their environmental conditions. A slight variation in environmental parameters creates a stressful condition that attenuates the immune system and increases the susceptibility of animals to microbial infection. 

#### 3.1.1. Temperature

Temperature is considered an important abiotic stressor, as a slight change in water temperature can affect the body physiology and health of crustaceans [105,106,107]. Interestingly, heat shock proteins are amongst the most significant proteins that are induced by hypo and hyperthermia and their role in protection against thermal stress has been well documented [108,109,110,111,112]. Among the Hsp multigenic family, few proteins are expressed at extremely low levels under normal conditions, while the transcription of most Hsps increased significantly in response to stresses, e.g., stress-inducible proteins (Hsp70). However, Hsps that are expressed constitutively under normal conditions, and may be upregulated during stress conditions, are generally known as heat shock cognate proteins, e.g., Hsc70 [113,114,115,116]. Hsps play a central role in thermotolerance by promoting growth at moderately high temperatures and protecting the organism from mortality at extremely high temperatures [117]. In general, Hsps are induced in both hyperthermia and hypothermia conditions. For example, an increase in the water temperature has been found to induce the production of Hsp70 in *Ferropenaeus chinensis*, Chinese white shrimp [118], *Gammarus pulex*, freshwater crustacean [119], and *A. franciscana*, [120]. Moreover, increased Hsp (Hsp90 and Hsp40) expression was also reported during 6 h of cold shock at 1 and 6 °C in *A. franciscana* [121]. Hsp70 has been demonstrated to play an important role in protecting cells from damage in *S. paramamosain* in response to thermal stress (this was shown in an increase in 11 °C from normal growth temperature) [122]. In addition, the adult *A. franciscana*, when exposed to sub-lethal heat shock (37 °C for 30 min), induces the transcription of Hsp70, Hsp67, and Hsc70, resulting in an improved tolerance of brine shrimp to high temperatures [123]. Interestingly, marine invertebrates are very sensitive to high temperatures and there are a myriad of reports suggesting that Hsp70 is upregulated in response to heat stress [124,125,126]. These studies indicate that Hsps improve thermal tolerance in crustaceans and provide protection in both hyperthermia and hypothermia conditions.

#### 3.1.2. Salinity

The concentration of dissolved inorganic salt concentrations or the salinity in the water is reported to affect the osmoregulation of crustaceans and induce cellular damage, including a deleterious effect on the folding and transformation of polypeptides [127,128,129,130]. Moreover, the osmotic-stress-induced Hsp production plays a crucial role in the maintenance of biological processes, as well as the protection of crustaceans against stressful conditions [131,132,133,134,135]. The study by Yang et al. [79] reported that high salinity stress increased the expression of Hsp70 and that this could lead to enhanced resistance in *S. paramamosain* against changes in water salinity. The possible mechanism behind the protective action of Hsps against salinity stress is that the osmotic change increases the metabolism rate and enhances the stress response resulting in increased Hsp production [136]. The increased Hsp response subsequently enhances the immune response, including lysozyme, phenoloxidase, and peroxide activity and provides rapid protection against osmotic stress until the organic osmolytes are fully accumulated [137,138]. Water salinity was reported as an important factor for the purposes of natural growth of crustaceans; further, studies on *P. trituberculatus* have shown that variable salinity significantly influences larval development [139]. Xu and Qin [61] found that Hsp60 has an important role in both the cellular and humoral stress response of the swimming crab, *P. trituberculatus*, and that these responses regulate the salinity stress via an intrinsic pathway. Further, these responses also play an essential role in protecting the swimming crab against salinity stress. In addition, a few reports have suggested that an increase in the expression of Hsp70 enhances immune response and confers protection to *A. franciscana* against hypersalinity stress [140]. 

#### 3.1.3. Environmental Pollutants

The environmental pollutants induced heat shock proteins (Hsps) expression in crustaceans are the most frequently studied in the literature [141,142,143]. The study of heat shock proteins in invertebrates started in the 1990s, and the first observation conducted on Hsps was made by Köhler et al. [144]. The study showed that the exposure of three diplopods (*Tachypodoiulus niger*, *Cylindroiulus punctatus*, and *Glomeris marginata*), one isopod (*Oniscus asellus*) and two slugs (*Arion ater* and *Deroceras reticulatum*) to heavy metals/molluscicides resulted in the increased expression of Hsp70 [145]. 

Studies on the effect of environmental pollutants on crustaceans demonstrated that the Hsp gene expression is induced by several chemical stresses. For example, this can be found in: nonylphenol (NP) (used in the polymer industry); bisphenol A diglycidyl ether (BPA) (intermediate in the production of polycarbonate and epoxy resins) [146]; 17α-ethynyl estradiol (EE) (synthetic estrogen) [147]; bis(2-ethylhexyl) phthalate (DEHP) (plasticizer in polymer products); endosulfan (ES) (organochlorine insecticide); chloropyriphos (CP) (organophosphorus insecticide); paraquat dichloride (PQ) (oxygen radical generating herbicide); Cadmium (Cd); lead (Pb) and potassium dichromate (Cr) (heavy metals); and benzo[a] pyrene (BaP) (polycyclic aromatic hydrocarbon) [148]. Moreover, for instance, the sublethal concentration of endosulfan has been reported to enhance the synthesis of Hsp70 and Hsp90 in monsoon river prawn, i.e., *M. malcolmsonii* and *P. monodon* [149,150,151,152]. In another study, the mixture of environmental pollutant chemicals has been reported to modulate the physiological, as well as immunity and survival responses in crustaceans. For instance, Park and Kwak [89] demonstrated that the application of bisphenol A (BPA) and 4-nonlphenol (NP)—an endocrine disrupting chemical (EDCs)—at different concentrations and at different time intervals (12, 24, 48, and 96 h) induce the expression of Hsp90; further, when exposed to BPA and NP, the marine crab, *Charybdis japonica*, has a significantly increased survival [54]. Additionally, chemical stress was reported to induce the production of Hsps, which helps in maintaining the homeostasis and structural integrity of cells [153,154]. 

### 3.2. Biotic Stresses

The heat shock proteins that are induced by biotic stresses play a very crucial role in protein folding, immune enhancement, and cross-protection against infectious diseases [155,156,157]. Although, there are several reports that demonstrate that Hsps are easily induced by abiotic stresses including heat, salinity, etc. [158], very little information is available on crustaceans’ Hsp response against biotic stresses, including bacteria, parasites, and viruses. Some of the recent research findings have suggested that members of the Hsp70 family have been identified in crustaceans, which are involved in the response to biotic stresses, mainly bacteria, parasites, and viruses [159,160]. Zhou et al. [119] demonstrated that *L. vannamei* when challenged with *V. alginolyticus* (Gram-negative) and *S. aureus* (Gram-positive) bacteria have a significantly increased expression of *L. vannamei* Hsp60 (LvHSP60) and Hsp70 (LvHSP70) gene in the gills, hepatopancreas, and hemocytes. In another study, temporal transcription of LvHSP70, following the white spot syndrome virus (WSSV) challenge, has been reported to induce an anti-WSSV innate immune response in *L. vannamei* [161]. 

In the swimming crab, *Portunus trituberculatus*, transcription of the *P. trituberculatus*, the Hsp70 (PtHsp70) gene was shown to increase very rapidly in response to the bacterial challenge with *V. alginolyticus* [162]. Similar findings were reported by Yang et al. [19], i.e., the fact that there was an increased transcription of Hsp70 in hemocytes of *S. paramamosain* after the *V. alginolyticus* challenge and were involved in generating cross-protection in the mud crab. Additionally, Hsp90, which plays a crucial role in protein biosynthesis, signal transduction, and immune responses, was also shown to induce protective immunity in crustaceans. Huang et al. [20] analyzed the role of Hsp90 in *S. paramamosain* in response to microbial infection, and the results showed that the transcription of SpHSP90 was upregulated in mud crabs after being challenged with *Staphylococcus aureus*, white spot syndrome virus (wssv), and *V. harveyi* [20]. 

Hsps exert their physiological effect via assisting in the formation of new polypeptides as well as in the protecting and maintaining of the host cell polypeptides and naïve proteins from denaturation during microbial infection [163,164,165]. In crustaceans, the host immune response against microbial infection [133] is often associated with reactive oxygen species (ROS) production [166]. It has been demonstrated that in *L. vannamei*, after bacterial challenge, the subsequent induction of the host immune response leads to an increase in ROS levels [167]. While microbial infection-induced ROS mostly has antimicrobial activity, the production of ROS can result in the denaturation of proteins (proteotoxicity) in the host cell itself; as such, in this condition, the induced Hsps display a cytoprotective role and act as chaperone proteins in order to maintain the protein homeostasis and preserve cellular structures [168]. 

## 4. An *A. franciscana* Model System to Investigate the Role of Hsps in Crustaceans

The *A. franciscana* is a small branchiopod crustacean that is highly osmotolerant and reported from several harsh environmental conditions worldwide [169,170,171,172]. They live in an environment of severe hypersalinity, high levels of ultraviolet radiation, fluctuating oxygen concentration, and extreme temperature [173,174]. The oviparous development in *A. franciscana* leads to the production of hard-shell covering diapause cysts, which are composed of stress-tolerant metabolically inactive embryos stalled at gastrulation and that which can remain in stasis for several years [175,176,177,178]. However, when the diapause cysts were immersed under appropriate conditions in seawater with aeration and temperatures, the hard-shell raptures and cysts develop, releasing swimming larvae within 24 h (Figure 4) [178,179,180]. 

Apart from its interesting life history, *A. franciscana* are non-selective filter feeders (which can be grown in a wide range of feed resources), have a rapid generation cycle (the cyst grows to adult in 20–30 days), require very low space for growth (hence have a relatively smaller cost to culture), and developmental stages are well characterized. This aspect of their stages being well characterized, for instance, is shown in the fact that the *A. franciscana* produces encysted gastrulae cysts during oviparous development, while ovoviviparous development provides live larvae in both sexual and asexual (parthenogenetic) stages. Additionally, gnotobiotic (germ-free) culture conditions (allowing full control over the host-associated microbial communities) and advanced molecular techniques including qPCR and RNAi are well established in *A. franciscana*, which makes this species an exceptional model organism that can be used in order to investigate the host–pathogen relationship and to study the biological activity of protective compounds [181,182,183,184,185,186,187,188,189,190]. In addition, cysts of *A. franciscana* can be stored for a couple of years in the fridge and, after terminating the diapause stage, the cysts from different generations can be used and hatched all together, simultaneously. This excellent facility of storage and hatching, permits one to perform experiments on demand, which can help to avoid or minimize environmental influences. Above all, the genome sequence of *A. franciscana* showed that it shares a very high homology with shrimps and other crustaceans’ genomes (Figure 5). Therefore, there is a high possibility that the outcome of studies based on *A. franciscana* would provide a fundamental basis to understand the host–pathogen interactions in other commercially important shrimp species. 

The induction of Hsp production inside the host in order to control diseases in aquaculture was investigated using the model organism, brine shrimp larvae [191,192]. In 1988, Miller and McLennan observed the presence of heat shock proteins in the early developmental stages of brine shrimp, i.e., in encysted gastrula embryos (cysts) and newly hatched nauplius larvae. They have reported that the larvae exhibited induced thermotolerance, which is associated with the synthesis and upregulation of heat shock proteins [193]. The role of heat shock protein response in adult *A. franciscana*, in response to high temperature (including LT50 determination, enhanced thermotolerance, and increased production of the Hsp70 family stress protein) was studied by Frankenberg et al. [95]. Results demonstrated that Hsp70 family proteins (mainly Hsp67 and Hsc70) levels were significantly upregulated during sublethal heat shock (37 °C for 30 min). The *A. franciscana* exposed to Zn-control/Cd-control treatment induced the expression of Hsp and increased the partitioning of Cd to the tropically available metal (TAM), which could result in the bio-enhancement of Cd trophic transfer to predators, which, in turn, leads to the suppression of Zn accumulation in *A. franciscana* [194]. 

Later, a gnotobiotic (germ-free) culture system was developed for brine shrimp and several studies have demonstrated that this provides a fully controlled and excellent host–pathogen environment and facilitates the determining of the effect of external stimuli on the host (Figure 5) [195]. The gnotobiotic system also avoids the interference generated by host-associated microorganisms as well as shifts in the composition of microbial diversity [196,197,198,199,200]. The effect of non-lethal heat shock (NLHS) on host survival, immune response, and protection against stressful conditions were also studied by several researchers. Results showed that NLHS increased the transcription of Hsp70, which enhances the immune response and provides cross-protection from environmental, physiological, and microbial stress in *A. franciscana* [201,202,203,204,205,206]. In another study, Sung et al. [64] demonstrated that non-pathogenic CAG 629 and CAG 626 *E. coli* strains, when heat shocked and administered through feed to *Artemia* larvae, resulted in enhanced Hsp70 expression (~two folds) and subsequent protection against *V. campbellii* infection [207]. In another study, the bacterial strains GR 8 (*Cytophaga* sp.), LVS 3 (*Aeromonas hydrophila*), LVS 2 (*Bacillus* sp.), LVS 8 (*Vibrio* sp.), and GR 10 (*Roseobacter* sp.) overproducing DnaK were reported to protect and improve gnotobiotic *Artemia* resistance to *V. campbellii* infection [208]. Baruah et al. [23] demonstrated that a heat shock protein inducer, Tex-OE^®^, enhances the production of Hsp70 and increased the survival of *A. franciscana* nauplii against thermal and salinity stresses. Further, it was, therefore, concluded that the protective effect of Tex-OE^®^ is mediated by an enhanced production of Hsp70 [209]. In addition, the production of heat shock protein 70 (Hsp70) in *A. franciscana* in response to non-lethal heat shock (30 min exposure to 37 °C) increased the tolerance of brine shrimp to zinc and cadmium metal exposure [210]. Later, several studies investigated the molecular chaperone activity of Hsps in *A. franciscana* through an in vivo RNA interference (RNAi) technique [211,212]. The results showed that the injection of dsRNA to *A. franciscana* (before fertilization) resulted in a complete knockdown of the expression of small heat shock proteins (i.e., p26) and reduced the resistance of cysts to desiccation and freezing [213,214]. In another study, the knockdown of dsRNA Hsp70 was reported to downregulate the expression of Hsp70 mRNA, thereby resulting in an reduced survival of shrimp and brine shrimp larvae against pathogenic *V. campbellii* and AHPND-causing *V. parahaemolyticus* challenges [215,216,217]. Taken together, these studies confirm that Hsps possibly play important roles in enhanced protection in crustaceans against stressful conditions by stabilizing and refolding unfolded or denatured proteins, as well as in enhancing their innate defense system. 

## 5. Heat Shock Protein, a Promising Candidate to Enhance Immunity and Prevent Diseases in Crustaceans

The role of Hsps, produced by both prokaryotic and eukaryotic organisms, in eliciting immune responses and inducing resistance to diseases has been well established in various animal and human models. However, in crustaceans, the studies into the effects of Hsps in generating protective immunity against infection stress are accumulating [218,219,220]. Based on these accumulating pieces of evidence, it can be suggested that Hsps are potential health-beneficial biomolecules that could be targeted in order to develop a disease-control strategy in aquaculture animals.

The Hsp70 family comprises the most well-characterized Hsps. The induction of Hsp expression could enhance crustacean immunity as shown in other animals that are exposed to heat stress. Shrimp (*L. vannamei*) that are exposed to chronic NLHS showed a higher expression of LvHSP70, LvHSP90, and immune-related genes (i.e., LvproPO and LvCrustin [221,222,223,224,225]). The high expression of LvproPO and hemocyanin was also observed in shrimp that were exposed to acute NLHS. Moreover, shrimp exposed to either acute or chronic NLHS had a higher survival rate than that of the non-heated shrimp control when they were challenged with VP_AHPND_. In addition, in regard to bacterial infection, the exposure of shrimp to NLHS could reduce WSSV infection as shown by the decreased viral copy number and the decreased cumulative mortality of WSSV-infected *L. vannamei*. In Artemia, it was also reported that NLHS could protect the animals against deleterious bacterial challenges. Furthermore, the functions of HSP70, HSC70, and HSC70-5 have been demonstrated to use the RNA interference (RNAi) technique. Iryani et al. [36] used RNAi to verify the role of HSP70 in protecting the nauplii of *A. franciscana* against abiotic and biotic stressors. The survival of nauplii lacking HSP70, compared with that of those with a functional HSP70, was decreased by 41% during heat stress and 34% upon *V. campbellii* infection. These results suggest that Hsps plays an important role in maintaining protein homeostasis by functioning as a molecular chaperone, while enhancing the host innate immune response against bacterial infection [226,227,228,229,230]. The effect of Hsps in regulating immune-related gene expression in crustaceans was not only investigated by suppression of HSP gene expression, but also by the injection of recombinant Hsps. In *L. vannamei*, injected with the recombinant DnaK followed by *V. campbellii* challenge, the transcript expression of immune genes TGase-1 and LvproPO-2 and endogenous HSP70 (LvHSP70) were clearly affected. Similarly, *P. monodon*, which was first injected with DnaK and then injected 1 h later with *V. harveyi*, showed a significant increase in proPO transcript expression. The study involving LvHSP70 injection showed that HSP70 could induce several immune pathways in the shrimp. Direct injection of rLvHSP70 into shrimp muscle demonstrated that rLvHSP70 enters hemocyte cells and localizes to both the cytoplasm and nucleus, while also accumulating in the plasma membrane. Thus, both gene knockdown and recombinant protein injection elicited similar results, suggesting a novel mechanism underlying the role of LvHSP70 in the activation of the shrimp immune system. Moreover, feeding *A. franciscana* with *E. coli* producing ArHsp70 or DnaK proteins showed a high survival rate in a *Vibrio* challenge assay. The observed effects could be due to the enhancement of the *Artemia* immune system as phenoloxidase activity was found to be increased by these proteins [231,232,233,234]. 

The Hsp family that is of main interest for disease control is HSP70, however the sHsps of HSP90 and HSP60, as well as the co-chaperone HSP40, appear to ameliorate infection by pathogens as well. sHsps provide oligomeric platforms for the ATP-independent binding of structurally perturbed proteins, preventing their irreversible denaturation when cells are stressed. HSP90, HSP70, and HSP60 are stress-induced and they have the ability to protect proteins from irreversible denaturation. However, the major function of these chaperone families is to bind and fold nascent proteins through ATP-driven allosteric rearrangement, although the molecular structure and mechanism of action for each chaperone differ. Hsps function cooperatively by forming intracellular networks of chaperones, co-chaperones, and accessory proteins. sHsp monomers, consisting of a conserved α-crystallin domain flanked by an amino-terminal sequence and a carboxyl-terminal extension [235,236,237,238], assemble into oligomers. The α-crystallin domain contributes to the dimerization of monomers and substrate binding, activities that depend on the amino- and carboxyl-terminal regions for the greatest efficiency [239,240,241]. Further, sHsp oligomers either disassemble or undergo structural rearrangement during stress, increasing surface hydrophobicity and enhancing reactions with substrate proteins [242,243,244,245,246]. Proteins released from sHsps when the stress passes, either refold spontaneously or refold with the assistance of an ATP-dependent Hsp, such as HSP70. The primary role of sHsps during exposure to stress, including infection, is to protect proteins from irreversible denaturation.

Under acute thermal stress in the culture environment, it was found that the white shrimp HSP40 (LvHSP40) transcript levels were significantly induced in muscle, gill, and the hepatopancreas. The expression profiles of four HSP genes (LvHSP60, LvHSP70, LvHSC70, and LvHSP90) of *L. vannamei* were significantly induced, and the transcription level of LvHSP70 was the most sensitive to temperature fluctuations. Moreover, HSP60 and HSP10 genes from *Scrippsiella trochoidea* were rapidly upregulated upon exposure to both low and high temperatures [247,248,249,250]. In pathological situations, such as necrotic cell death, Hsps can be released into the extracellular environment with cellular proteins in order to induce autoimmunity by receptor-mediated activation of the innate immune response. In shrimp, there is evidence that the Hsps are highly expressed in response to pathogen infection. The LvHsp60 protein from *L. vannamei* was significantly upregulated in the gills, hepatopancreas, and in the hemocytes after being challenged with either Gram-positive and Gram-negative bacteria. 

In addition, it has been shown that using a plant-based polyphenolic compound, phloroglucinol, induces HSP70 production and protects the brine shrimp *A. franciscana* and freshwater prawn *M. rosenbergii* against bacterial infection. Treatment of the brine shrimp with phloroglucinol was shown to result in significant upregulation of DSCAM, proPO, Peroxinectin, HSP90, and HSP70 and downregulation of LGBP. Moreover, the phenolic compound carvacrol also induces the expression of HSP72 in *A. franciscana*. Further, the induction of HSP72 enhances Artemia larvae tolerance to lethal heat stress or pathogenic *V. harveyi*. Similarly, treating *A. franciscana* with pyrogallol at an optimum concentration could induce protective effects against *V. harveyi* infection [40,41,178,179,180,181,182]. Taken together, these findings suggest that Hsps possibly contribute to immune defense against infections by regulating the immune system in crustaceans.

## 6. Conclusions and Future Perspective

Aquaculture is a rapidly growing food-producing sector. The sector has grown at an average rate of 8.9% per year since 1970, compared to only 1.2% for capture fisheries and 2.8% for terrestrial farmed meat-production systems over the same period. However, the intensive development of the aquaculture industry has been accompanied by an increase in environmental impacts. The production process generates substantial amounts of polluted effluent, containing uneaten feed, and feces. Discharges from aquaculture into the aquatic environments contain nutrients, as well as various organic and inorganic compounds such as ammonium, phosphorus, dissolved organic carbon, and organic matter. The high levels of nutrients cause environmental deterioration of the receiving water bodies. In addition, the drained water may increase the occurrence of pathogenic microorganisms and introduce invading pathogen species. Additionally, the degraded aquatic environment makes the animal more susceptible to disease outbreaks. For instance, over the last decade of disease outbreaks, in particular, bacterial diseases have brought socio-economic and environmental unsustainability to the crustacean aquaculture industry. As estimated by the FAO, the economic losses from a disease outbreak in the aquaculture industry are over USD 9 billion per year, which is approximately 15% of the value of world-farmed fish and shellfish production. The traditional methods applied so far in the mitigation of stressful conditions, such as disinfectants and antibiotics, have had very limited success. Among the various drugs, antibiotics are the most popular and widely used chemotherapeutic agent used to control bacterial disease in crustacean aquaculture. However, the wide use of antibiotics, and especially the application at subtherapeutic doses, can eliminate both pathogenic and beneficial bacteria and inevitably results in the development of resistant strains of antibiotics that can create environmental problems in the ecosystems. In addition, the presence of antibiotic residues in commercialized products of aquaculture constitutes an additional problem for human health, as this can generate problems of allergy, toxicity, and results in the alteration of the human gut’s normal microflora. Hence, the development of natural products or plant-derived compounds is needed in order to mitigate stressful conditions in aquatic environments and enhance the immune reactivity of crustaceans. 

Interestingly, to avoid the problems generated by environmental change, crustaceans possess their own specific behavioral and physiological adaptive mechanism, i.e., the enhanced heat shock protein response, that plays a significant role by stabilizing the protein structure and function, as well as generating protective immunity against abiotic and biotic stresses. However, when the adverse environmental condition is prolonged, it surpasses the natural defense capacity and affects growth, survival, metabolism, reproduction, and immunity. Hence, the management strategies, e.g., the use of heat shock protein-inducing (Hspi) compounds may offer a promising viable and sustainable means to maintain homeostasis in aquatic animals and avoid integrative and or multiple stresses. Natural products from medicinal plants and marine seaweeds are considered potential alternatives for the prevention of stressful conditions in crustacean aquaculture. Plant-based compounds are identified in order to possess the characteristic of enhancing the heat shock protein within the animal in a non-invasive manner. These compounds/molecules are also commonly called heat shock protein inducers (Hspi). Heat shock proteins, especially Hsp70, are reported to be functionally involved in providing cross-protection in crustacean species. For instance, the upregulation of Hsp70 production in shrimps as a general stress response protects the animals from subsequent secondary heterologous environmental and physiological insults. In general, the molecular chaperone activity—which maintains protein homeostasis by protecting the nascent polypeptides from misfolding, assisting in the assembly and disassembly of macromolecular complexes, as well as facilitating co- and post-translational folding and regulating translocation—is documented as a protective function of Hsp70. Additionally, Hsp70 is also reported to induce thermotolerance, protect against osmotic stress, prevent oxidative toxicity and damage, and improve tolerance against microbial infection. These observations clearly illustrated that Hsps play a significant role in host immunity and health. In conclusion, we can say that the development of a compound or molecule that could have a possible application as a Hsp inducer would be a holistic approach that could be used to generate tolerance in the crustacean species against subsequent deleterious environmental stresses. In addition, it may be a suitable candidate for use as an anti-stress agent in crustacean aquaculture. 

## Figures and Tables

**Figure 1 life-12-01777-f001:**
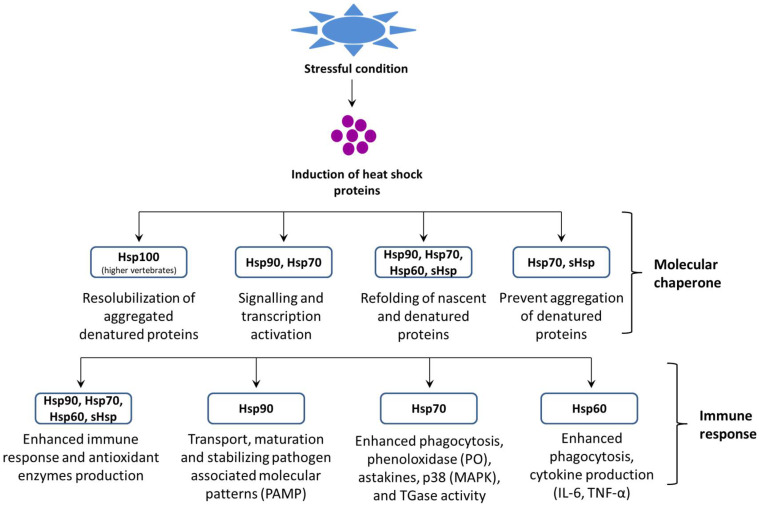
Role of heat shock proteins (Hsps) in proteostasis and the host immune response.

**Figure 2 life-12-01777-f002:**
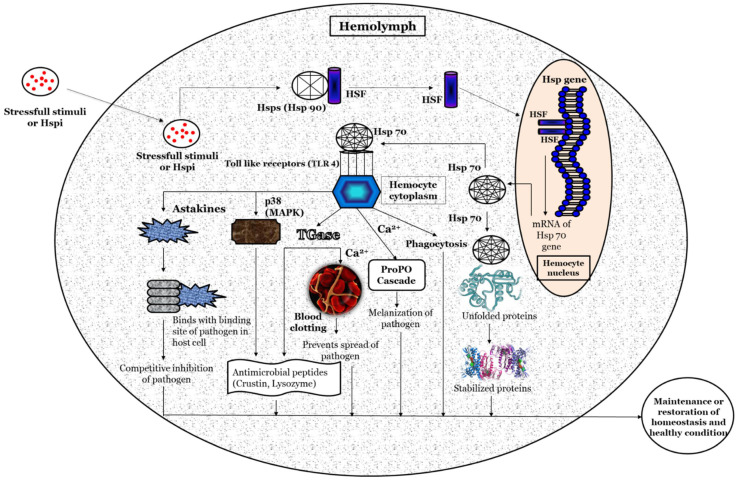
A schematic hypothetical illustration of the possible mechanism of action for Heat shock proteins (Hsps) in crustaceans.

**Figure 3 life-12-01777-f003:**
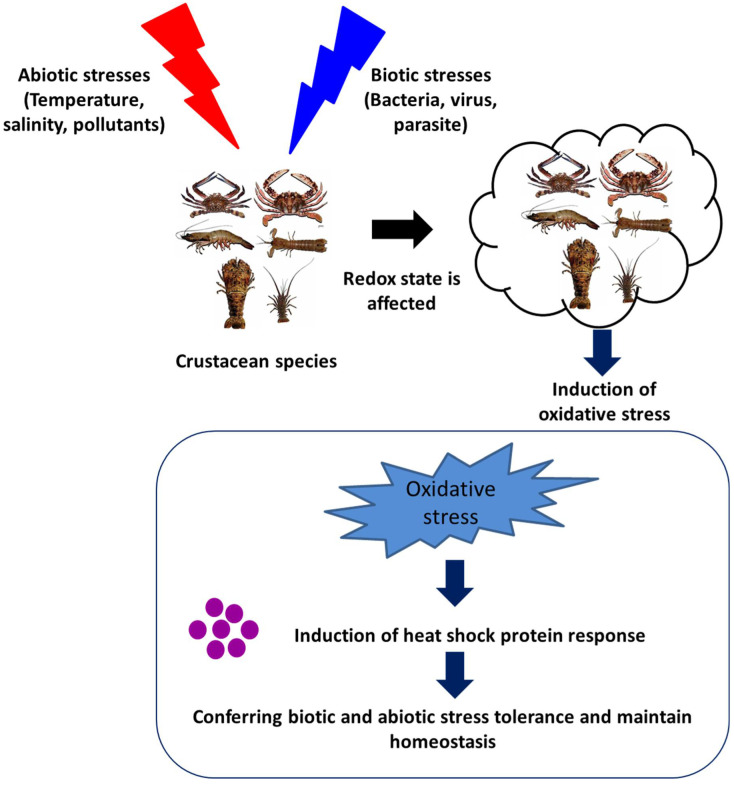
Effect of abiotic and biotic stresses on host health and heat shock protein response.

**Figure 4 life-12-01777-f004:**
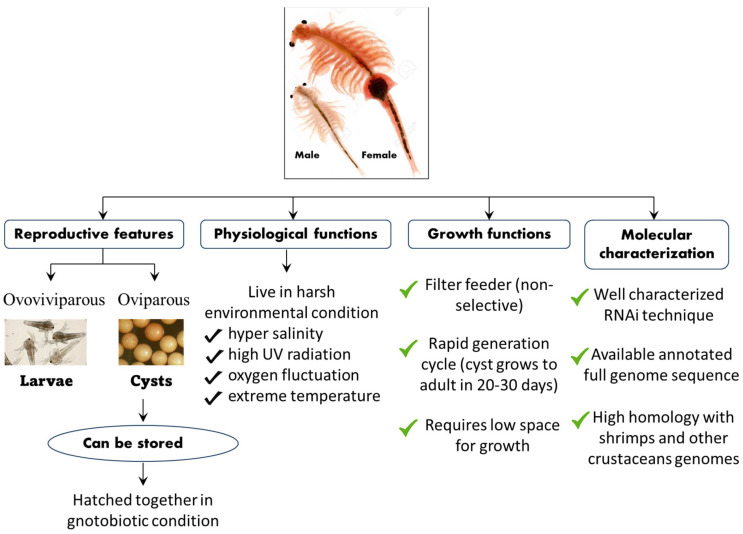
Schematic representation of the life cycle, physiological, growth, and molecular features of the *A. franciscana* model system.

**Figure 5 life-12-01777-f005:**
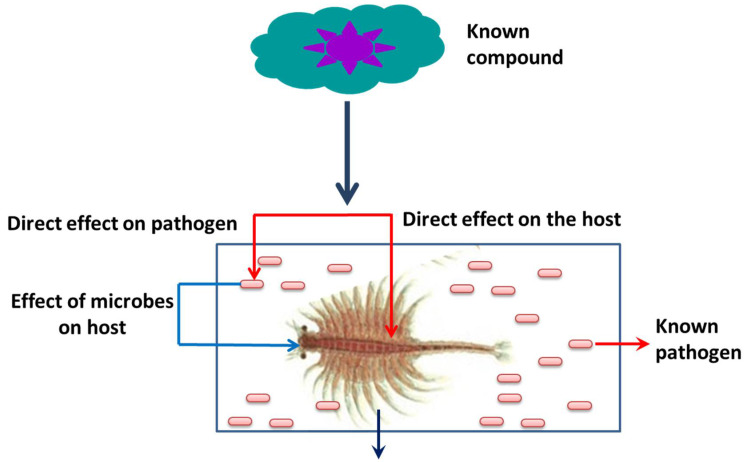
Advantage of a gnotobiotic (germ free) *A. franciscana* model system.

**Table 1 life-12-01777-t001:** Factors that modulate heat shock protein production and their functional significance in aquatic invertebrates.

Species	Hsp Inducing Condition	Dosage & Duration	Hsp Induced	Tissue Examined	Immune Response	Disease/Stress Resistance	References
*Artemia franciscana* (Brine shrimp)	Temperature shock	30 min sub-lethal heat shock at 37 °C	Hsp70	Whole animal (adult)	--	High temperature (+)	[94]
		30 min heat shock from 21 °C to 37 °C	Hsp70	Whole animal (adult)	--	High temperature (+)	[95]
		30 min NLHS at 37 °C, 6 h recovery	Hsp70	Whole animal (nauplii)	--	High temperature (+)	[96]
		30 min heat shock from 28 °C to 37 °C, 6 h recovery	Hsp70	Whole animal (nauplii)	--	*Vibrio campbellii* and *V. proteolyticus* (+)	[24]
		1 h cold shock from 28 °C to 4 °C and heat shock from 28 °C to 37 °C, 6 h recovery	Hsp70	Whole animal (nauplii)	--	High temperature and *Vibrio campbellii* (+)	[21]
	Feeding Hsp overproducing bacteria	Feeding *Escherichia coli* overproducing prokaryotic Hsp (Dnak)	Dnak	Whole animal (nauplii)	--	*Vibrio campbellii* (+)	[97]
		Feeding *Escherichia coli* strain (YS2 and A native) overproducing *Artemia* Hsp 70 and Dnak	Hsp70, Dnak	Whole animal (nauplii)	proPO (+)	*Vibrio campbellii* (+)	[78]
	Feeding Hsps	Feeding truncated portion of Hsp 70	Hsp70, Dnak	Whole animal (nauplii)	proPO (+)	*Vibrio campbellii* (+)	[98]
	Plant-derived/natural compounds	Feeding Tex-OE^®^ (Hspi compound) with 20 µL/L to 160 µL/L concentration	Hsp70	Whole animal (nauplii)	---	*Vibrio campbellii* (+)	[28]
		Pretreatment Pro-Tex^®^ (Hspi compound) with 152 ppb for 1 h	Hsp70	Whole animal (nauplii)	--	High temperature and hypersalinity (+)	[99]
		Feeding Tex-OE^®^ (Hspi compound) with 2.5 mg/L to 50 mg/L concentration	Hsp70	Whole animal (nauplii)	proPO and TGase (+)	*Vibrio campbellii* and *V. harveyi* (+)	[27]
		Feeding phenolic pyrogallol (Hspi compound) with 79 µM to 1185 µM concentration	Hsp70	Whole animal (nauplii)	proPO and TGase (+)	*Vibrio harveyi* (+)	[25]
		Feeding poly-β-hydroxybutyrate (PHB) with 10 mg/L to 1000 mg/L concentration	Hsp70	Whole animal (nauplii)	proPO and TGase (+)	*Vibrio campbellii* (+)	[26]
		Phloroglucinol pretreatment (Hspi compound) with 30 µM concentration	Hsp70	Whole animal (nauplii)	--	*Vibrio parahaemolyticus* (AHPND strain) (+)	[100]
		Phloroglucinol treatment (Hspi compound) with 2 µM concentration	Hsp70	Whole animal (nauplii)	DSCAM, proPO, PXN, Hsp90, Hsp70, and LGBP (+)	*V. parahaemolyticus* (AHPND strain), *V. harveyi* and high temperature (+)	[9]
		Sodium ascorbate pretreatment (Hspi compound) with 200 ppm concentration	Hsp70	Whole animal (nauplii)	SOD and GST (+)	*Vibrio harveyi* (+)	[101]
*Penaeus monodon* (Tiger prawn)	Temperature shock	4 days of exposure to either 0.1 µg L^−1^ or 1 µg L^−1^	Hsp90	Muscle		Pesticide (endosulfan and deltamethrin) (+)	[102]
		1 h heat shock from 26 to 37 °C, 30 min recovery	Hsc70	Hemocytes	--	High temperature (+)	[87]
		24 h heat shock from 29 °C to 35 °C	Hsp70	Tail muscle	--	Gill-associated virus (+)	[71]
*Litopenaeus vannamei* (Pacific white shrimp)	Administration of beneficial bacteria	Injection of 10 µL (1 × 10^7^ cells mL^−1^) of live *V. alginolyticus*	Hsp60, Hsp70	Hemocytes, muscle, stomach, heart, gill, and hepatopancreas	LvHSP60 and LvHSP70 (+)	--	[103]
*Macrobrachium rosenbergii* (Freshwater prawn)	Feeding plant-derived/natural compounds	Phloroglucinol treatment (Hspi compound) with 5–10 µM concentration	Hsp70	Whole animal (nauplii)	--	*Vibrio parahaemolyticus* (AHPND strain) (+)	[104]
*Portunus trituberculatus* (Japanese blue crab)	Osmotic stress	25 ppt to 10 ppt and 40 ppt for 24 h	Hsp60	Gill, gill muscle, ovary, antennal gland, abdominal muscle, hypodermis, heart, and intestine	PtHsp60	Osmotic stress (+)	[76]
*Scylla paramamosain* (Mud crab)	Administration of beneficial bacteria	Crab injected with 20 µL live *V. alginolyticus* (10^7^ cells/mL)	Hsp70	Midgut, stomach, hepatopancreases, epidermis, thoracic ganglion, gill, eyestalk, heart, brain, muscles, and hemocytes	--	*Vibrio alginolyticus*, osmotic stress, and high temperature (+)	[19]
	Administration of beneficial bacteria	Crabs were injected with 50 µL *S. aureus* (5 × 10^7^ CFU), 50 µL *V. harveyi* (5 × 10^7^ CFU), and 50 µL WSSV supernatant (5 × 10^5^ virus particles)	Hsp90	Hepatopancreases, stomach, gill, intestine, muscle, connective tissue, gonad, heart, and hemocytes		*Staphylococcus aureus*, *Vibrio harveyi*, and WSSV (+)	[20]
	Osmotic stress	15 ppt to 30 ppt for 96 h	Hsp70	Midgut, stomach, hepatopancreases, epidermis, thoracic ganglion, gill, eyestalk, heart, brain, muscles, and hemocytes	--	Osmotic stress (+)	[19]
	Temperature shock	96 h of heat shock from 25 °C to 36 °C and cold shock from 25 °C to 10 °C	Hsp70	Midgut, stomach, hepatopancreases, epidermis, thoracic ganglion, gill, eyestalk, heart, brain, muscles, and hemocytes	--	High temperature (+)	[19]

(+): positive effect; (--): Not studied; NLHS: non-lethal heat shock; Hspi: heat shock protein inducing compound; NP: nonylphenol; BPA: bisphenol A diglycidyl ether; EE: 17a-ethynyl estradiol; DEHP: bis(2-ethylhexyl) phthalate; proPO: Prophenoloxidase; TGase: transglutaminase; DSCAM: Down syndrome cell adhesion molecule; PXN: peroxinectin; LGBP: lipopolysaccharide and β-1,3-glucan-binding protein; SOD: Superoxidase dismutase; GST: Glutathione S-transferase; LvHSP70: *L. vannamei* heat shock protein 70; LvHSP60: *L. vannamei* heat shock protein 60; PtHsp60: *P. trituberculatus* heat shock protein 60; and WSSV: white spot syndrome virus.

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
