# Peer review of "Heat Shock Proteins (Hsps) in Cellular Homeostasis: A Promising Tool for Health Management in Crustacean Aquaculture"

_life, 2022, doi:10.3390/life12111777_

Round 1
Reviewer 1 Report
This review covers detailed aspects of heat shock proteins in aquatic organisms, with some reference to crustaceans. The paper is well referenced and the authors have gathered a lot of data. It is clear that English is not the native language of the authors. Therefore the manuscript would benefit from a thorough edit by a native English speaker.
I found the title somewhat misleading. I was looking for information as to how heat shock proteins could be used to improve yields, reduce stress, or be used as a health indicator in crustaceans, but this was lacking. Instead there were essentially 8 pages that provided a general introduction to heat shock proteins in a wide array of species. This could be reduced substantially. The next 12 pages do detail heat shock proteins in crustaceans and this does provide useful information. That being said, it is really a list of induction of heat shock proteins by various environmental and anthropogenic stressors. There is then another section (functional importance) which appears more general and only somewhat focused on crustaceans.
Then after over 600 lines of text comes the section for which the paper is titled, but this is barely a page long and it doesn’t go into any real detail. Then only half a page discusses the potential heat shock inducers – with no real detail into how these work, or how they could be incorporated in aquaculture.
Overall the manuscript does produce a synopsis of heat shock proteins, but these kind of reviews can be found elsewhere. Either the title of the review needs to be changed, or better still, reduce most if the background information and concentrate more on specifically how heat shock proteins and inducers thereof can be of use to the aquaculture industry.
This review covers detailed aspects of heat shock proteins in aquatic organisms, with some reference to crustaceans. The paper is well referenced and the authors have gathered a lot of data. It is clear that English is not the native language of the authors. Therefore the manuscript would benefit from a thorough edit by a native English speaker.
I found the title somewhat misleading. I was looking for information as to how heat shock proteins could be used to improve yields, reduce stress, or be used as a health indicator in crustaceans, but this was lacking. Instead there were essentially 8 pages that provided a general introduction to heat shock proteins in a wide array of species. This could be reduced substantially. The next 12 pages do detail heat shock proteins in crustaceans and this does provide useful information. That being said, it is really a list of induction of heat shock proteins by various environmental and anthropogenic stressors. There is then another section (functional importance) which appears more general and only somewhat focused on crustaceans.
Then after over 600 lines of text comes the section for which the paper is titled, but this is barely a page long and it doesn’t go into any real detail. Then only half a page discusses the potential heat shock inducers – with no real detail into how these work, or how they could be incorporated in aquaculture.
Overall the manuscript does produce a synopsis of heat shock proteins, but these kind of reviews can be found elsewhere. Either the title of the review needs to be changed, or better still, reduce most if the background information and concentrate more on specifically how heat shock proteins and inducers thereof can be of use to the aquaculture industry.
Author Response
- I found the title somewhat misleading. I was looking for information as to how heat shock proteins could be used to improve yields, reduce stress, or be used as a health indicator in crustaceans, but this was lacking. Instead there were essentially 8 pages that provided a general introduction to heat shock proteins in a wide array of species. This could be reduced substantially. The next 12 pages do detail heat shock proteins in crustaceans and this does provide useful information. That being said, it is really a list of induction of heat shock proteins by various environmental and anthropogenic stressors. There is then another section (functional importance) which appears more general and only somewhat focused on crustaceans.
- As suggested by the reviewer, the first 8 pages were extensively checked and reduced in the revised manuscript.
Then after over 600 lines of text comes the section for which the paper is titled, but this is barely a page long and it doesn’t go into any real detail. Then only half a page discusses the potential heat shock inducers – with no real detail into how these work, or how they could be incorporated in aquaculture.
- We appreciate and agree to the reviewer comments. The entire manuscript is checked to incorporate more information about ole Hsps in crustaceans aquaculture.
- Overall the manuscript does produce a synopsis of heat shock proteins, but these kind of reviews can be found elsewhere. Either the title of the review needs to be changed, or better still, reduce most if the background information and concentrate more on specifically how heat shock proteins and inducers thereof can be of use to the aquaculture industry.
- As suggested by the reviewer, we have tried to modify the entire manuscript to present an updated and more informative manuscript.
Reviewer 2 Report
Eeditorial revisions will be needed.
1. English name (with scientific name) should be indicated when it appeared for the first time, thereafter, only English name or scientific name should be used.
2. hsp(s), HSP(s), Hsp for heat shock protein are used randomly. Need to use only one consistently throughout the manuscript.
3. Delete a space between value and % throughout the text
4. line 8: add (hsps, Hsps or HSPs) after Heat shock proteins
5. line 79: V. should be spelled out
6. line 86: Artemia franciscana (brine shrimp) should be A . franciscana or brine shrimp
7. line 92: The heat shock proteins (hsps) should be Hsps or HSPs
8. line 106: (1974) should be [41]
9. line 107: Drosophila should be D.
10. line 121: (1986) should be [63]
11. line 131: sHsps, Dnak, ssa, ssb, ssc, ssd should be spelled out here
12. lines 149, 150, 176: are should be is ?
13. line 159: EEVD should be spelled out
14. line 171: role should be roles ?
15. line 178: Staphylococcus should be italic
16. lines 197,205, 199, 204: brine shrimp, mud crab should be used
17. line 202: L. should be spelled out
18. line 203-204: Homarus americanus (lobster) should be American lobster
19. line 236: L. vannanei (Whiteleg shrimp) should be Whiteleg shrimp
20. line 270: (2018) should be [160]
21. line 273: (2018) should be [6]
22. line 275: P. monodon should be tiger shrimp
23. lines 291, 292: brine shrimp, tiger shrimp should be used
24. line 299: heat shock element (HSE) should be HSE
25. lines 309, 310, 311: only HSF, HSP, HSE should be used
26. line 338: (Sung et al., 2011) should be [38]
27. lines 339, 349: HSP, HSPs ?
28. lines 352, 256, 358: hsps or HSPs or Hsps? need consistency
29. lines 361, 364, 366: only use English names
30. line 389: (2013) should be [19]
31. line 390: only use English name
32. line 399: Xu and Qin (2012) has not been found in References
33. line 401: (1992) should be [95]
34. line 426: (2014) should be [5]
35. line 428: hours should be h
36. line 440: (2010) should be [133]
37. line 448: (2013) should be [19]
38. lines 449, 453, 469, 473, 479, 482, 487, 489, 493, 494, 496, 505, 509, 512, 528, 537, 544, 592, 601, 603, 604, 609, 610: brine shrimp
39. line 465: (Fu et al. 2011) should be [124]
40. line 495: (Vos et al. 2014) should be [221]
41. line 501: add [224] after McLennan
42. lines 504. 560, 569, 571, 586, 589, 653, 677, 688, 690: hsp, HSP or Hsp
43. line 507: (2000) should be [109]
44. line 529: (2009) should be [78]
45. line 535: (2012b) should be [227]
46. lines 589, 595: mud crab
47. lines 599, 653, 682: heat shock protein inducing (Hspi) should be Hspi
48. line 601: (2012, 2014), (2014) should be [23, 27], [28]
49. line 611: (2016) should be [243]
50. line 664: hSPs ?
51. line 689: in vivo should be italic
52. References: titles should be small capital: ref.no. 53, 111, 122, 146, 233, 242, 252, 255, 259, 266, 267, 268
53. lines 851, 924, 927, 1078: delete (80-.)
54. line 943: check the journal name
55. lines 1047, 1200, 1231, 1265-1266, 1268, 1301, 1307, 1320, 1322: scientific names of animals should be italic
56. line 1093: add USA
57: Ref no. 183: is this a book or journal?
58: lines 1303, 1305, 1309: pages are missing
Author Response
- 1. English name (with scientific name) should be indicated when it appeared for the first time, thereafter, only English name or scientific name should be used.
- Modified in the revised manuscript
- hsp(s), HSP(s), Hsp for heat shock protein are used randomly. Need to use only one consistently throughout the manuscript.
- Modified in the revised manuscript
- Delete a space between value and % throughout the text
- Modified in the revised manuscript
- line 8: add (hsps, Hsps or HSPs) after Heat shock proteins
- Modified in the revised manuscript
- line 79: V. should be spelled out
- Modified in the revised manuscript
- line 86: Artemia franciscana (brine shrimp) should be A . franciscana or brine shrimp
- Modified in the revised manuscript
- line 92: The heat shock proteins (hsps) should be Hsps or HSPs
- Modified in the revised manuscript
- line 106: (1974) should be [41]
- Modified in the revised manuscript
- line 107: Drosophila should be D.
- Modified in the revised manuscript
- line 121: (1986) should be [63]
- Modified in the revised manuscript
- line 131: sHsps, Dnak, ssa, ssb, ssc, ssd should be spelled out here
- Modified in the revised manuscript
- lines 149, 150, 176: are should be is ?
- Modified in the revised manuscript
- line 159: EEVD should be spelled out
- Modified in the revised manuscript
- line 171: role should be roles ?
- Modified in the revised manuscript
- line 178: Staphylococcus should be italic
- Modified in the revised manuscript
- lines 197,205, 199, 204: brine shrimp, mud crab should be used
- Modified in the revised manuscript
- line 202: L. should be spelled out
- Modified in the revised manuscript
- line 203-204: Homarus americanus (lobster) should be American lobster
- Modified in the revised manuscript
- line 236: L. vannanei (Whiteleg shrimp) should be Whiteleg shrimp
- Modified in the revised manuscript
- line 270: (2018) should be [160]
- Modified in the revised manuscript
- line 273: (2018) should be [6]
- Modified in the revised manuscript
- line 275: P. monodon should be tiger shrimp
- Modified in the revised manuscript
- lines 291, 292: brine shrimp, tiger shrimp should be used
- Modified in the revised manuscript
- line 299: heat shock element (HSE) should be HSE
- Modified in the revised manuscript
- lines 309, 310, 311: only HSF, HSP, HSE should be used
- Modified in the revised manuscript
- line 338: (Sung et al., 2011) should be [38]
- Modified in the revised manuscript
- lines 339, 349: HSP, HSPs ?
- Modified in the revised manuscript
- lines 352, 256, 358: hsps or HSPs or Hsps? need consistency
- Modified in the revised manuscript
- lines 361, 364, 366: only use English names
- Modified in the revised manuscript
- line 389: (2013) should be [19]
- Modified in the revised manuscript
- line 390: only use English name
- Modified in the revised manuscript
- line 399: Xu and Qin (2012) has not been found in References
- Modified in the revised manuscript
- line 401: (1992) should be [95]
- Modified in the revised manuscript
- line 426: (2014) should be [5]
- Modified in the revised manuscript
- line 428: hours should be h
- Modified in the revised manuscript
- line 440: (2010) should be [133]
- Modified in the revised manuscript
- line 448: (2013) should be [19]
- Modified in the revised manuscript
- lines 449, 453, 469, 473, 479, 482, 487, 489, 493, 494, 496, 505, 509, 512, 528, 537, 544, 592, 601, 603, 604, 609, 610: brine shrimp
- Modified in the revised manuscript
- line 465: (Fu et al. 2011) should be [124]
- Modified in the revised manuscript
- line 495: (Vos et al. 2014) should be [221]
- Modified in the revised manuscript
- line 501: add [224] after McLennan
- Modified in the revised manuscript
- lines 504. 560, 569, 571, 586, 589, 653, 677, 688, 690: hsp, HSP or Hsp
- Modified in the revised manuscript
- line 507: (2000) should be [109]
- Modified in the revised manuscript
- line 529: (2009) should be [78]
- Modified in the revised manuscript
- line 535: (2012b) should be [227]
- Modified in the revised manuscript
- lines 589, 595: mud crab
- Modified in the revised manuscript
- lines 599, 653, 682: heat shock protein inducing (Hspi) should be Hspi
- Modified in the revised manuscript
- line 601: (2012, 2014), (2014) should be [23, 27], [28]
- Modified in the revised manuscript
- line 611: (2016) should be [243]
- Modified in the revised manuscript
- line 664: hSPs ?
- Modified in the revised manuscript
- line 689: in vivo should be italic
- Modified in the revised manuscript
- References: titles should be small capital: ref.no. 53, 111, 122, 146, 233, 242, 252, 255, 259, 266, 267, 268
- Modified in the revised manuscript
- lines 851, 924, 927, 1078: delete (80-.)
- Modified in the revised manuscript
- line 943: check the journal name
- Modified in the revised manuscript
- lines 1047, 1200, 1231, 1265-1266, 1268, 1301, 1307, 1320, 1322: scientific names of animals should be italic
- Modified in the revised manuscript
- line 1093: add USA
- Modified in the revised manuscript
57: Ref no. 183: is this a book or journal?
- Modified in the revised manuscript
58: lines 1303, 1305, 1309: pages are missing
- Modified in the revised manuscript
Round 2
Reviewer 1 Report
This version is much improved upon the first. However, I still found it hard going and it was not until the last section that I was again reminded that this was primarily (supposed to be) connected to aquacultured crustaceans. At present it still leads like a list of crustacean hsp responses. I think the authors need to remind the readers about the focus of the review. The last section should be much earlier in the paper, even in the introduction. Tell the reader how important crustacean aquaculture is, how much is is growing, what are the potential problems farmers will face and how can crustaceans adapt to certain conditions. At present it still reads as a list about hsp - without fully putting it in the context of the importance in aquaculture (which is in the title)
For example in the section on temperature effects - give us more detail about temperature in aquaculture - that each species has an optimal growth zone. When and where does temperature vary in aquaculture operations and by how much. What devastating effects can this have on production. And then tell us how hsp can help the animal, or protect it from detrimental effects. Same with salinity, when and where does this vary in an aquaculture operation and how will animal responses protect them

Author Response
We sincerely thank you for your response and comments.
We have tried to incorporate and revised the manuscript as per the reviewer comments.
I sincerely hope that the revised version of the manuscript will meet the scientific rigor, and journal standard and will be considered by Life journal.